# Multimodal false information detection method based on Text-CNN and SE module

**Yi Liang** [1,2], **Turdi Tohti**[1,2]*, **Askar Hamdulla**[1,2]

**1** School of Information Science and Engineering, Xinjiang University, Urumqi, China, **2** Xinjiang Key Laboratory of Signal Detection and Processing, Urumqi, China

* turdy@xju.edu.cn

**Data Availability Statement:** Twitter dataset is available from https://github.com/MKLab-ITI/image-verification-corpus Weibo dataset is available from https://doi.org/10.1145/3123266.3123454.

## Abstract

False information detection can detect false information in social media and reduce its negative impact on society. With the development of multimedia, the multimodal content contained in false information is increasing, so it is important to use multimodal features to detect false information. This paper mainly uses information from two modalities, text and image. The features extracted by the backbone network are not further processed in the previous work, and the problems of noise and information loss in the process of fusing multimodal features are ignored. This paper proposes a false information detection method based on Text-CNN and SE modules. We use Text-CNN to process the text and image features extracted by BERT and Swin-transformer to enhance the quality of the features. In addition, we use the modified SE module to fuse text and image features and reduce the noise in the fusion process. Meanwhile, we draw on the idea of residual networks to reduce information loss in the fusion process by concatenating the original features with the fused features. Our model improves accuracy by 6.5% and 2.0% on the Weibo dataset and Twitter dataset compared to the attention based multimodal factorized bilinear pooling. The comparative experimental results show that the proposed model can improve the accuracy of false information detection. The results of ablation experiments further demonstrate the effectiveness of each module in our model.

## Introduction

With the development of information technology, social media has become the main way for people to obtain information, especially during the epidemic, people's lives are more closely connected with social media. The rapid development of social media not only brings convenience to people, but also facilitates the spread of false news and misleading news. False information is defined as unsubstantiated stories and statements [1]. The spread of false information can mislead the public and have a negative impact on society. For example, the 2012 Doomsday Theory, which declares that the Earth will experience a major catastrophe on December 21, 2012, or "three consecutive days of darkness", this rumor has caused panic among people around the world, causing people to spend a lot of money to hoard shopping supplies, and even spend a lot of money to build "Noah's Ark".

**Funding:** National Natural Science Foundation of China(62166042). National Natural Science Foundation of China(U2003207). Natural Science Foundation of Xinjiang, China (2021D01C076). Strengthening Plan of National Defense Science and Technology Foundation of China (2021-JCJQ-JJ-0059). The funders had no role in study design, data collection and analysis, decision to publish, or preparation of the manuscript.

**Competing interests:** The authors have declared that no competing interests exist.

Fig 1 presents several multimodal false information posts from the Twitter dataset [2]. Each post contains a paragraph of text and an image associated with it. The image in the first post has been altered so the image is false and the text is false, the second post image is real but the image is about the Sicily air disaster. In the final post, the image was edited to include a shark that didn't exist during Hurricane Sandy. The dissemination of this false information has a serious impact on the normal operation of society, so it is very important to find out how to detect false information and stop it from spreading. For false information containing text and images, can be divided into three categories. The first type of false information is that the text content is false but the image is true, the second type of false information is that the text content is true but the image is false, and the third type of false information is that both text content and images are false.

In recent years, deep learning models have been used for false information detection, early approaches focused on detection using text features [3, 4], such as the model proposed by Pérez [5] uses textual content to detect false information, but this model can only detect the first and third types of false information, but cannot correctly detect the second type of false information, if we use both text and image information at the same time, all kinds of false information can be detected [6–8], which reflects the importance of multi-modal false information detection.

There are two challenging problems in existing research work. First, how to extract higher-quality text features and image features. Second, how to better fuse text features and image features to obtain more valuable fusion features. Previous works have used RNN (Recurrent Neural Network) [9] or Transformer-based models [10] to extract text features, CNN (Convolutional Neural Network)-based [11] models to extract image features, and finally fusing text and image features through simple splicing, factorization bilinear pool or attention mechanism. However, these methods directly fuse the features extracted from the backbone network, and fail to perform corresponding processing on the extracted features to make up for the insufficiency of the features extracted from the backbone network in some aspects. Furthermore, these research works do not consider the problems of noise and information loss in the feature fusion process. This paper proposes a false information detection model based on Text-CNN [12] and SE (Squeeze-and-Excitation Networks) [13] modules, which solves the above problems well. Our model uses 3 scales of Text-CNN to make up for the slight deficiency

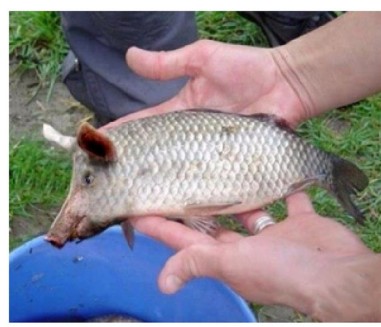

Text: New Species of Fish found in Brazil or just Really good Photoshop

(a)

Text: Missing airplane malaysia airlines mh370

(b)

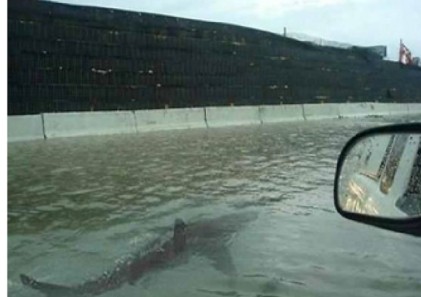

Text: So crazy that is literally a shark swimming in someones front yard in new jersey hurricane sandy

(c)

**Fig 1. Example of fake news in the Twitter dataset.**

of the Transformer-based model in the extraction of local features, at the same time, we adopt our modified SE module to reduce the influence of noise in the fusion process, and reduce the information loss in the fusion process by concatenating the original features and the fusion features. Therefore, our model can better detect false information.

The main contributions of this paper are as follows:

1. We use 3 different scales of Text-CNN to process the text features extracted by the pre-trained model BERT [14] and the image features extracted by the pre-trained model SWTR [15] to obtain more valuable features.

2. We modify the SE module so that it can fuse text features and image features. We utilize the channel attention mechanism in the SE module to mitigate the effect of noise during fusion to obtain better-represented fused features.

3. We draw on the idea of residual network to concatenate original features and fusion features to reduce the information loss in the fusion process.

4. The accuracy and F1 value of our model on Twitter dataset and Weibo dataset [16] outperform the baseline model AMFB.

## Related work

Traditional false information detection models are mostly text-based. In earlier studies, people mainly extracted text features manually. Qazvinian et al. [17]. exploited n-grams, bi-grams features extracted from text to detect rumors. Pérez et al. [5]. extracted five linguistic features from the text to detect error messages. With the development of technology, researchers found that the artificial extraction of features will be limited by the dataset, resulting in the extracted features not having generality [18]. Subsequent researchers used deep learning techniques to allow computers to automatically extract features from text to detect false information. Liu et al. [19]. proposed a model that uses CNN to extract text features and detect false information. The model uses CNN to mine deeper text features that humans cannot discover. Ma et al. [20]. used RNN to extract text features, and the model used RNN to discover content related to textual contextual content. Nasir [21] utilizes both CNN and RNN to extract text features, which can combine the advantages of CNN and RNN.

In recent years, with the increasing number of forms of information expression, how to use different forms of information simultaneously to detect false information detection has attracted the attention of many researchers. Two forms of information, text and images, are often used in existing research work to detect false information. Singhal et al. [22]. The text and image features extracted by BERT and VGG19 are concatenated and fed into the classifier to obtain detection results. Kumari et al. [23]. proposed a multi-modal fusion model based on multi-modal factorization bilinear pooling, the model first uses a combination of BILSTM and Attention to extract text features, followed by a combination of CNN, BiGRU and Attention to extract image features, and finally the two features are fused by Multimodal Factorized Bilinear Pooling and fed into the detector to obtain detection results. Song et al. [24]. proposed a multi-modal false information detection model based on cross-modal attention residuals and multi-channel CNN. The model can extract information related to the target modality from the remaining modalities without losing the information of the target modality, while the influence of noise during the fusion of information from different modalities can be reduced by multi-channel CNN. Dhawan [25] proposed a multi-modality detection model based on a graph neural network, which can allow fine-grained interactions within and between different modalities to further improve the accuracy of multi-modal false information detection. Wu

et al. [26]. proposed a novel multimodal co-attention network to better fuse text and image features for false information detection. With the rise of pre-trained models, researchers have conducted research on fusion algorithms that fuse text features and image features. Xu et al. [27]. divided the existing Transformer—based pre-trained fusion models into six categories, (1) Early Summation [28, 29], the model takes text and image features, weights them together and feeds them into the Transformer layer to fuse text and image features. This fusion method does not increase the computational complexity, but requires manual setting of the weights. (2) Early Concatenation [30–33], this model concatenates text features and image features and then inputs them into the Transformer layer to fuse the features of different modalities. This approach increases computational complexity. (3) Multi-stream to One-stream [34], this model inputs text features and image features into two Transformer layers for processing, and then concatenates them through another Transformer layer to fuse features. (4) One-stream to Multi-stream [35], this model concatenates text features and image features and inputs them into the Transformer layer for fusion, and then divides the fused features into two parts and inputs them into two different Transformer layers. (5) Cross-Attention [36, 37], using two Transformer layers to process text features and image features by exchanging two Q(Query) to complete the fusion of text features and image features. (6) Cross-Attention to Concatenation [38, 39], the text features and image features processed by Cross-Attention are concatenated and input to another Transformer layer for processing.

In addition to text and image information, other forms of information can also be used to detect false information. Wang [40] and others found that the existing research ignores the role of the strong emotion of the image in the rumor content, proposes a multimodal rumor detection model composed of visual emotion and textual emotion. Azri [41] proposed an end-to-end model that utilizes three features of text, images and emotion simultaneously. Armin [42] proposed a multimodal detection model that supports the fusion of different levels and types of information, which can simultaneously utilize textual, visual, user reviews and metadata.

## Multimodal false information detection method based on Text-CNN and SE module

Problem Definition: Suppose $P = \{p_1, p_2, \cdots, p_m\}$ each post in this dataset contains data in the form of both text and images, where $p_i$ represents the ith post. $T_{set} = \{t_1, t_2, \cdots, t_m\}$ is the text set, $t_i$ represents the text content in the ith post, $V_{set} = \{v_1, v_2, \cdots, v_m\}$ is the image set, $v_i$ represents the image contained in the $i$th post, $L = \{l_1, l_2, \cdots, l_m\}$ is the tag set, $l_i$ is the tag of the $i$th post, $p_i = \{t_i, v_i, l_i\}$. The main purpose of disinformation detection is to find a function $f(T, V) = Y$, this function identifies the authenticity of a post by the text information and image information in the post. $Y = \{y_1, y_2, \cdots, y_m\}$, $y_i$ is the predicted label of the ith post.

The model in this paper mainly consists of four parts: text feature extraction, image feature extraction, image and text feature fusion and classifier. Fig 2 shows our proposed multimodal false information detection method based on Text-CNN and SE module.

The model first uses BERT to extract the features of each token from the text, and concatenates the features of each token as text features, and its dimension is (33/95,768). Use SWTR to extract image features with dimension (49,768), then fuses text features and image features through modified SE module to obtain fused features, and text and image features are processed by Text-CNN with widths of 1, 2 and 3 to improve feature quality. Then the fused features, locally enhanced text features and image features are concatenated and fed into a classifier to classify them.

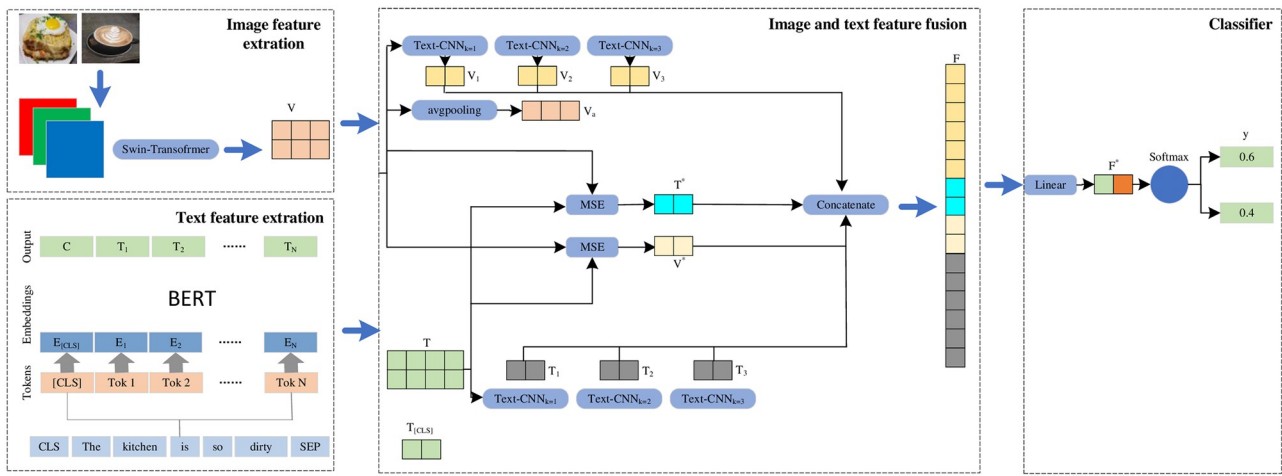

**Fig 2. Multimodal false information detection method based on Text-CNN and SE module.**

## Text feature extraction

Usually, when people post on social media, they express their thoughts in the form of text. The text contains the main meaning that the publisher wants to express. Therefore, how to process the text and extract high-quality text features has a significant impact on the detection accuracy of the model. This paper extracts text features through the combination of BERT and Text-CNN models.

BERT is a Transformer-based pre-training model. BERT is first trained on a large unsupervised dataset to learn some general knowledge, and then the learned knowledge is transferred to a specific task. BERT, due to its special structure, can achieve good results while reducing the consumption of training resources. However, the extraction of local features by BERT is slightly inadequate, so this paper uses Text-CNN to process the text features extracted by BERT, so that the text features contain more local information. Text-CNN is a CNN model applied to text. The convolution kernel of Text-CNN will keep the length of the convolution kernel consistent with the length of the word feature, and only adjust the width of the convolution kernel. The model can extract similar features to *n*-grams. The calculation of text features extracted using BERT is as follows:

$$PE = Position\_Embeddings(t) \tag{1}$$

$$SE = Segment\_Embeddings(t) \tag{2}$$

$$TE = Token\_Embeddings(t) \tag{3}$$

$$T_{all} = PE + SE + TE \tag{4}$$

*Position_Embeddings*() is a function that encodes the position of the text, *Segment_Embedding*() is a function that encodes the paragraphs of the text, and *Token_Embeddings* () is a function that transforms each word in the text into a word vector, where t is a set of text, $T_{all} = \{T_{[cls]}, T\}$ is the text feature extracted by BERT, $T \in R^{n*d_i}$, $T_{[cls]} \in R^{d_i}$, $n$ is the number of words in a sentence, and di is the dimension of each token feature vector. In this paper, we process $T$ by three

scales of Text-CNN, and the calculation formula is as follows:

$$T_1 = \varphi(conv1(W_1, T) + b_1)_{k=1} \tag{5}$$

$$T_2 = \varphi(conv1(W_2, T) + b_2)_{k=2} \tag{6}$$

$$T_3 = \varphi(conv1(W_3, T) + b_3)_{k=3} \tag{7}$$

$\varphi$ is the activation function, $conv1()$ is the function of the one-dimensional convolution operation, $W_1$, $W_2$, $W_3$ are convolution kernels that can be obtained by learning, and $b_1$, $b_2$, $b_3$ are bias that can be obtained by learning, and $k$ represents the width of the convolution kernel. $T_1$, $T_2$, $T_3 \in R^{64}$ are the text features obtained after three widths of Text-CNN processing.

## Image feature extraction

Images are more believable than text content, so an accurate image feature extraction module plays an important role in false information detection models. We use SWTR to extract image features and further process the extracted image features through Text-CNN with three widths.

SWTR is a successful model for using Transofrmer in computer vision. SWTR can extract both local features and global features through a windowing mechanism compared to CNN-based models, and SWTR's unique windowing mechanism reduces computational effort compared to the rest of Transformer-based models. SWTR has SOTA performance on multiple tasks. Since SWTR and BERT are both Transformer-based models, this model is therefore also similar to the BERT model in that it has some shortcomings in the treatment of local features. Since the windowing mechanism of SWTR cannot set the size of the convolutional kernel as flexibly as CNN, this paper uses Text-CNN to process the image features extracted by SWTR. In this paper, the image features extracted by SWTR are input into three different widths of Text-CNN for processing. The specific calculation process is as follows:

$$V = SWTR(v) \tag{8}$$

$$V_1 = \varphi(conv1(W_4, V) + b_4)_{k=1} \tag{9}$$

$$V_2 = \varphi(conv1(W_5, V) + b_5)_{k=2} \tag{10}$$

$$V_3 = \varphi(conv1(W_6, V) + b_6)_{k=3} \tag{11}$$

$SWTR()$ is the function of extracting image features using Swin-Transformer, $\varphi$ is the activation function, conv1 is the function for the 1D convolution operation, $W_4$, $W_5$, $W_6$ are the learnable convolution kernels, $b_4$, $b_5$, $b_6$ are the biases, $k$ is the scale of the convolution kernel, $v$ represents an image in the post, $V \in R^{z*d_v}$ is the image feature, $z$ is the number of extracted features, and dv is the dimension of the feature vector. $V_1$, $V_2$, $V_3 \in R^{64}$ are the image features after processing by Text-CNN of three widths.

## Feature fusion

So far we have the text features $T_{all}$ extracted by BERT, the image features V extracted by SWTR, text features $T_1$, $T_2$, $T_3$ and image features $V_1$, $V_2$, $V_3$. The SE module is mostly used for channel feature enhancement of the input feature maps in computer vision tasks. For example, if we input a feature map A with dimensions (H,W,C), the SE module will input A into two

full connection layers to get the attention score, and then multiply the feature map A with the attention score in the dimension of the channel to get the final output. We modify the SE module so that it can fuse text features and image features to obtain multimodal fusion features. The MSE module is shown in Fig 3. The SE module can assign weights to each channel,

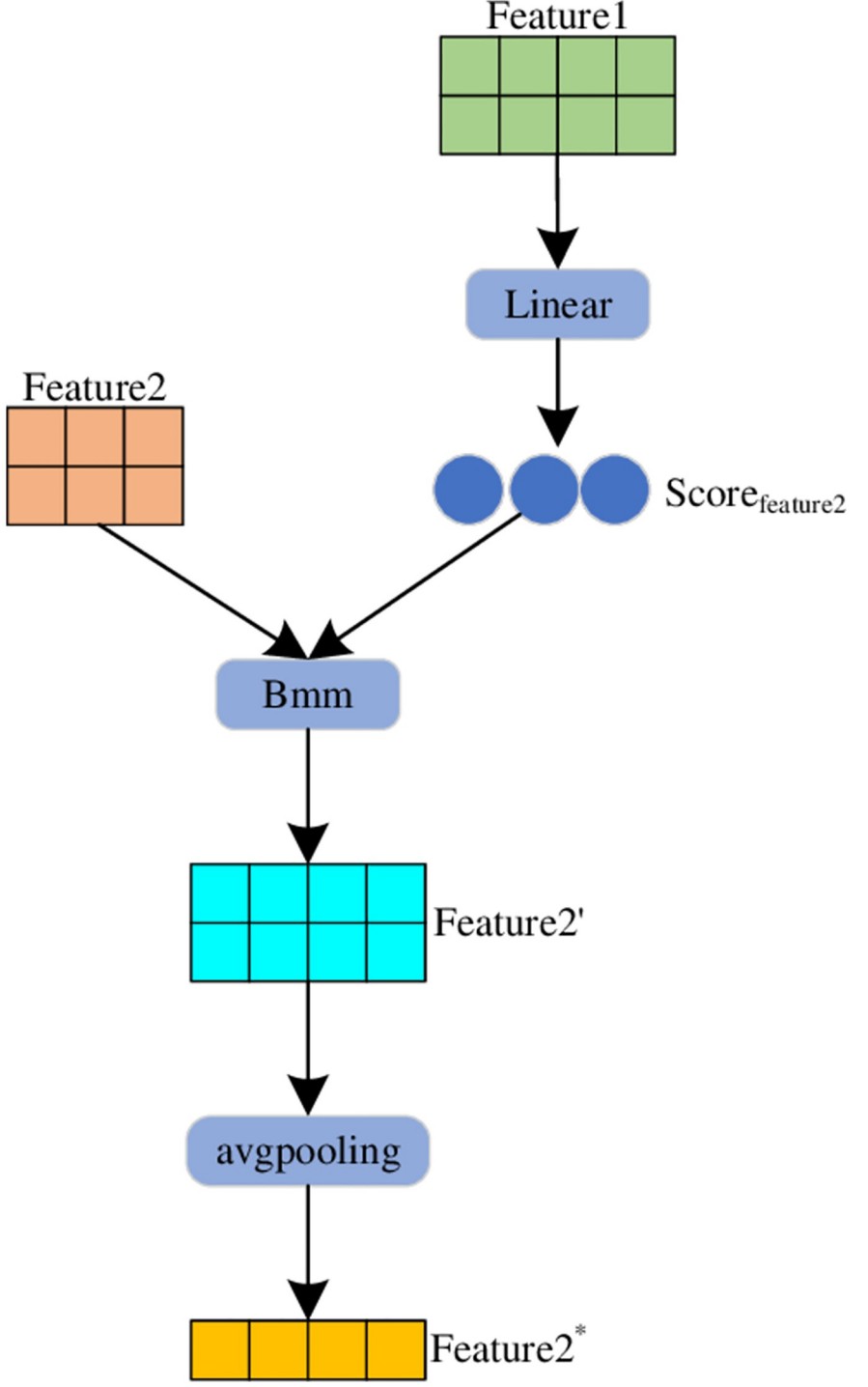

**Fig 3. Multimodal fusion of SE modules (MSE).**

automatically filter low-weight noise points. With a small increase in the number of parameters, the performance of the model on related tasks can be greatly increased. Therefore, we modify the SE module to fuse text and image features. The calculation process is as follows:

$$Score_{text} = Linear(V) \tag{12}$$

$$T' = Bmm(Score_{text}, T) \tag{13}$$

$$T^* = avgpooling(\beta(T')) \tag{14}$$

$$Score_{image} = Linear(T) \tag{15}$$

$$V' = Bmm(Score_{image}, V) \tag{16}$$

$$V^* = avgpooling(\beta(V')) \tag{17}$$

In this paper, the attention scores between different modalities are obtained through the fully connected layer. $Score_{image} \in R^{n^*z}$, $Score_{text} \in R^{z^*n}$ are the attention scores of text features on image features and the attention scores of image features on text features. $Bmm()$ is the function that performs the dot product operation. $T^* \in R^n$, $V^* \in R^z$ are image features and text features fused by the MSE module, $\beta$ is the activation function. Then we concatenate the text features and image features processed by Text-CNN and the fusion features extracted by the MSE module. The calculation process is as follows:

$$F = concatenate(T_1, T_2, T_3, V_1, V_2, V_3, T^*, V^*) \tag{18}$$

$Concatenate()$ is a function of concatenation operation, $F \in R^{64^*6+n+z}$ is the fusion feature that is finally input into the classifier.

## False information detection

We feed the fused features into fully connected layer and Softmax layer to obtain detection results.

$$F^* = Linear(F) \tag{19}$$

$$p_i = Softmax(F^*) \tag{20}$$

$$y_i = \arg\max(p_i) \tag{21}$$

We use the cross-entropy loss function to calculate the loss value:

$$L = -\sum_{i=1}^{m}[l_i \log p_i + (1 - l_i)\log(1 - p_i)] \tag{22}$$

$p_i$ is the probability that the post is false, argmax is the function of select the predicted label values, $y_i$ is the predicted label value of the post by the model, $m$ is the number of posts, and $l_i \in \{0, 1\}$ is the true label value, 1 represents false information, and 0 represents true information.

## Experiment and analysis

### Dataset and experimental settings

Machine configuration and environment for this experiment: CPU: Intel Xeon E5-2630L v3, 62 G memory, 8cores, GPU: NVIDIA GeForce RTX 3090, PyTorch(1.7.1), Python(3.8), Cuda

**Table 1. Data distribution of the Weibo and Twitter datasets.**

| Dataset | Train | | Test | | Image |
|---|---|---|---|---|---|
| | False | Real | False | Real | |
| Twitter | 6827 | 4993 | 717 | 1215 | 410 |
| Weibo | 6476 | 4096 | 1136 | 1215 | 13274 |

(10.2). To compare with previous work, we use the Twitter and Weibo datasets to complete our experiments. These are two publicly available, high quality datasets that can be used for multimodal disinformation detection.

The Twitter dataset is published by Boididou et al. The dataset contains training dataset and test dataset. The training dataset contains three types of information: false, true and humorous, but the test dataset contains only two types of information: true and false, so we remove the humorous type of information from the training dataset. The Weibo dataset is a multimodal Chinese dataset that contains only two types of posts, real and false. We split the posts containing multiple images in the Twitter dataset and the Weibo dataset into multiple posts containing only one image, and deleted the data containing only images, only text and images as gifs and black and white images. Table 1 shows the data distribution for the Weibo and Twitter datasets.

The two datasets above are publicly available datasets applied to false information detection studies. We are only interested in the text, images and labels in the dataset, so some remaining information is removed. We take text and images as input to the model and labels as facts. First we preprocess the text and images, for the text part we remove punctuation, URL and emoticons in the sentence, and for the image part,we resize all the images to (224, 224, 3). The training dataset is used to train the model, and the test dataset is used to verify the performance of the model.

Table 2 lists all the hyperparameters used to train the model.

## Comparative experiment

We implement some uni-modal and multimodal models to verify the validity of our model.
Uni-modal based models:

- BERT: We use $T_{[cls]}$ extracted from the fine-tuned BERT-Base as text features. The text features $T_{[cls]}$ are fed into the classifier to detect the authenticity of posts.

- SWTR: We use the image feature $V$ extracted from the SWTR model to feed into the average pooling layer to obtain the image features $V_a$, and then input the processed image feature into the classifier to detect the authenticity of posts.

**Table 2. Hyperparameters used in model training.**

| Parameters | Twitter | Weibo |
|---|---|---|
| Text length | 33 | 95 |
| Image size | (224,224,3) | (224,224,3) |
| Batch size | 70 | 70 |
| Optimizer | Adam(lr = 0.0001) | Adam(lr = 0.00005) |
| Epochs | 100 | 100 |
| Dropout | 0.2 | 0.6 |

Multimodal based models:

- att-RNN [16]: att-RNN is an RNN with attention mechanism that fuses text and image features for false information detection.

- EANN [43]: EANN (Even Adversarial Neural Network, EANN) is an end-to-end event adversarial network that uses an event discriminator to remove the impact of event information on detection results and improve the generality of the model.

- MVAE [44]: MVAE (Multimodal Variational Autoencoder, MVAE) is used to learn the correlation between modalities and then combined with a classifier to detect false information.

- AMFB [23]: AMFB (Attention based multimodal Factorized Bilinear, AMFB), the network uses BILSTM and VGG19 to extract text and image features, finally uses multimodal decomposition bilinear pooling to fuse features of text and images.

In order to verify the effectiveness of our proposed model, we compare the above baseline model with our model on both Weibo and Twitter datasets. At the same time, we conduct 5 experiments for each model under the same experimental conditions and take the average of the 5 experiments as the final result, the aim of which is to reduce the influence of experimental errors on the experimental results. The experimental results are shown in Table 3. According to the accuracy and F1 value, our model has better performance than the existing baseline models. It can also be observed that the performance of the single-modal model is lower than the multimodal model on both datasets, this suggests that using both text and image information can be more effective in detecting false information.

Figs 4 and 5 show the accuracy and loss values of our model when trained on the Twitter and Weibo datasets. The full form of 'iter' is 'iterations'. As can be seen from the figure, the loss gradually decreases to an equilibrium position, followed by a slight fluctuation at the equilibrium position, which indicates that the model is learning properly. It can be observed from the figure that our model is fully trained on both datasets, while our model has more difficulty in achieving convergence on the Weibo dataset because the Weibo dataset contains more images and different posts from different events, while most of the posts on the Twitter dataset are from the same event.

**Table 3. Comparative results for the Weibo and Twitter datasets.**

| Dataset | Model | Accuracy | F1 value of False News | F1 value of Real News |
|---------|-------|----------|------------------------|-----------------------|
| Twitter | BERT | 0.831 | 0.852 | 0.802 |
|         | SWTR | 0.834 | 0.871 | 0.766 |
|         | att-RNN | 0.664 | 0.676 | 0.651 |
|         | EANN | 0.741 | 0.610 | 0.810 |
|         | MVAE | 0.745 | 0.758 | 0.730 |
|         | AMFB | 0.883 | 0.920 | 0.810 |
|         | OUR | 0.903 | 0.924 | 0.866 |
| Weibo | BERT | 0.870 | 0.880 | 0.858 |
|       | SWTR | 0.713 | 0.702 | 0.724 |
|       | att-RNN | 0.772 | 0.692 | 0.754 |
|       | EANN | 0.791 | 0.780 | 0.80 |
|       | MVAE | 0.824 | 0.809 | 0.837 |
|       | AMFB | 0.832 | 0.840 | 0.830 |
|       | OUR | 0.897 | 0.902 | 0.890 |

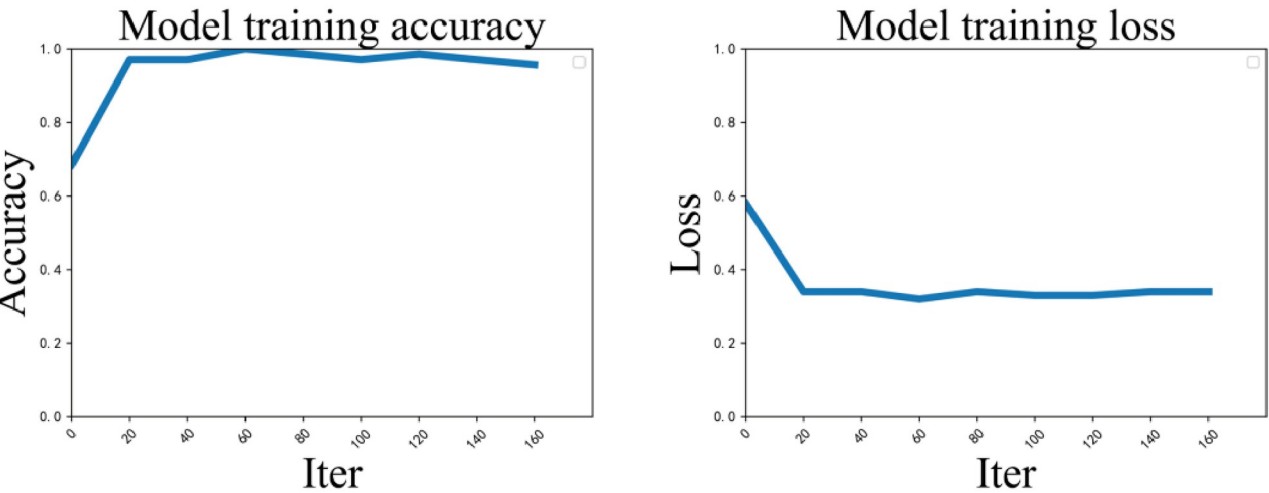

**Fig 4. Accuracy and loss curves of the model when trained on the Twitter dataset.**

### Ablation experiment

We set up 4 ablation experiments to demonstrate the effectiveness of our model.

- Ablation Experiment 1: We compare our model with the original model after removing different modules to demonstrate the validity of each module.

- Ablation Experiment 2: We compare the original BERT and SWTR models with our improved BERT and SWTR models to demonstrate the validity of our improvements.

- Ablation Experiment 3: We set up a series of experiments to demonstrate that the model is most effective in processing text and image features using three different scales of Text-CNNs simultaneously.

- Ablation Experiment 4: Several different fusion methods were used to demonstrate the effectiveness of the fusion methods we used by comparing them with the fusion methods we used.

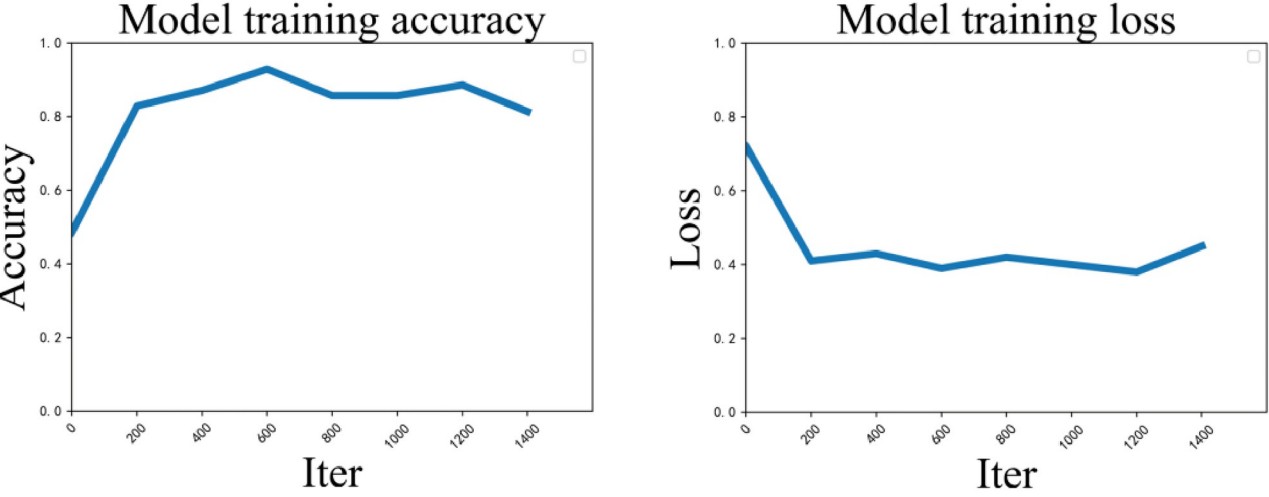

**Fig 5. Accuracy and loss curves of the model when trained on the Weibo dataset.**

**Table 4. Comparative results of ablation experiments on the Weibo dataset 1.**

| Model | Accuracy | F1 value of False News | F1 value of Real News |
|---|---|---|---|
| •MSE-Text-CNN | 0.887 | 0.894 | 0.879 |
| •MSE | 0.893 | 0.900 | 0.885 |
| •Text -CNN | 0.889 | 0.896 | 0.881 |
| OUR | 0.897 | 0.902 | 0.890 |

For the above 4 groups of ablation experiments, in order to eliminate errors, we performed 5 experiments for each model and took the average value.

**Ablation experiment one.** To demonstrate the effectiveness of each module in our proposed model, we conduct ablation experiments and the results are shown in Table 4:

- OUR: The complete model presented in this paper.

- -SE: We simply splice the text features $T_1$, $T_2$, $T_3$ extracted by BERTcnn and the image features $V_1$, $V_2$, $V_3$ extracted by SWTRcnn to detect the authenticity of the post. But remove MSE module.

- -Text-CNN: We use the MSE module to fuse text features $T$ and image features $V$, excluding text features and image features processed by Text-CNN.

- -SE-Text-CNN: We simply concatenate the text feature $T_{[cls]}$ extracted by BERT-Base and the image feature $V_a$ processed by the average pooling layer and input it into the classifier to detect the authenticity of the post. Removed the MSE module and Text-CNN module.

From Table 4, the complete model achieves the best results, demonstrating the effectiveness of each module. We can observe that the model with any module removed shows a decrease in accuracy compared to OUR. -SE dropped by 0.4%, -Text-CNN dropped by 0.8%, -SE-Text-CNN dropped by 1.0%. The MSE module in the model can alleviate the problem of noise introduced in the fusion process, so that the model can better fuse text and image features. It also reduces information loss during fusion by simply concatenating text features and image features that have been processed by Text-CNN.

**Ablation experiment two.** To verify that our improvements to BERT and SWTR are effective. We compare the original BERT and SWTR with our improved BERT and SWTR, and the results are shown in Fig 6.

- BERT: We input the text features $T_{[cls]}$ extracted by BERT into the classifier to get the detection result of the post.

- SWTR: We input the image features $V_a$ obtained from the averaging pooling layer into the classifier to obtain the classification results of the post.

- $BERT_{cnn}$: We obtained features $T_1$, $T_2$ and $T_3$ by processing the text features T extracted by BERT with three different scales of Text-CNN. Subsequently, $T_1$, $T_2$ and $T_3$ are concatenated and fed into the classifier to obtain detection results.

- $SWTR_{cnn}$: We use three different scales of Text-CNN to process the image features $V$ to obtain features $V_1$, $V_2$ and $V_3$, which are subsequently concatenated and fed into the classifier to obtain detection results.

As can be seen in Table 4, our improvements to BERT and SWTR are effective. The BERTcnn model improved the accuracy of the Weibo dataset by 1.1% compared with the

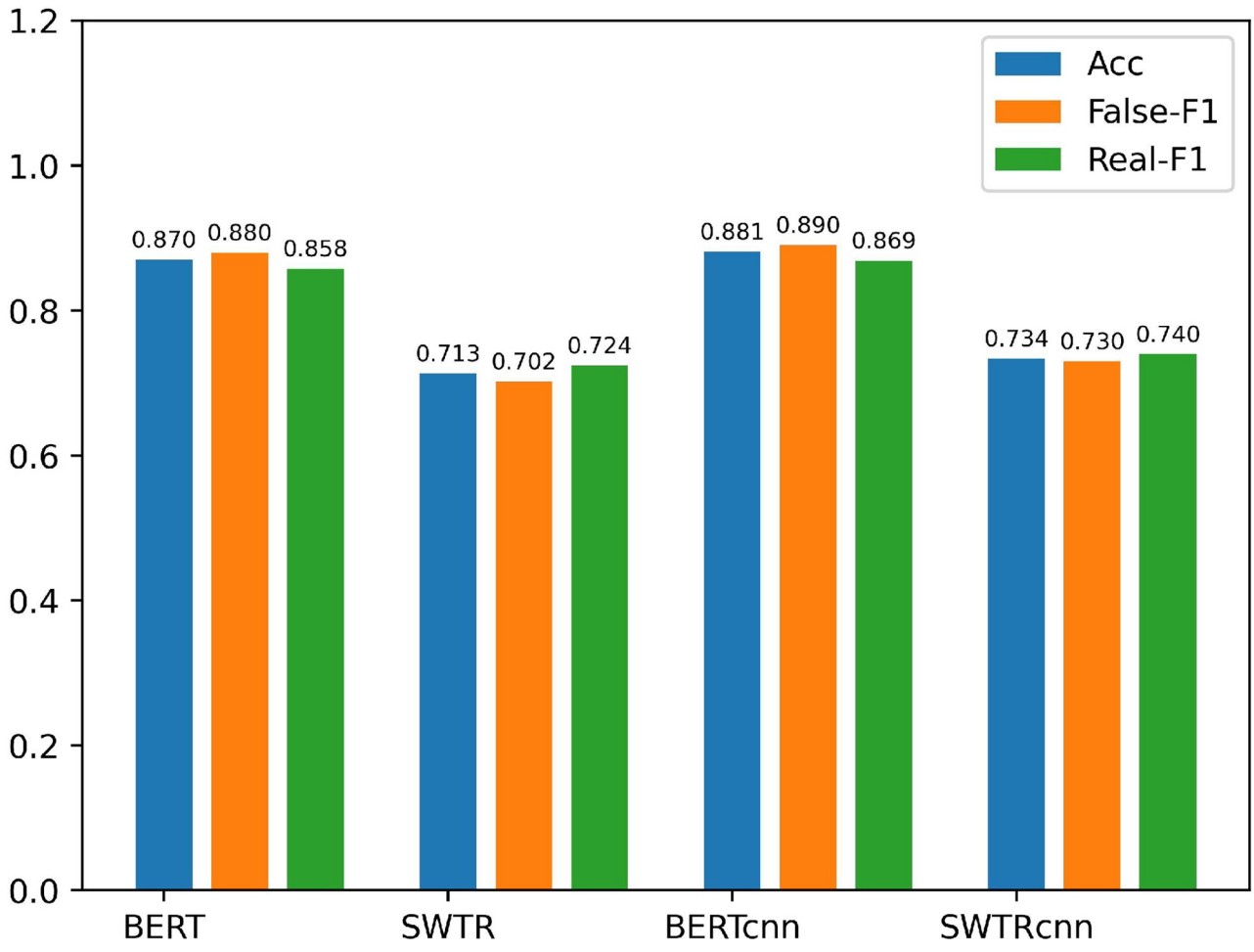

**Fig 6. Experimental results of ablation experiment 2.**

BERT, and the SWTRcnn model improved the accuracy of the Weibo dataset by 1.6% compared with the SWTR model. The experimental results prove our conjecture that the text and image features extracted by the Transformer-based pre-trained model can be further improved by Text-CNN processing.

**Ablation experiment three.** In order to demonstrate the effectiveness of using three different scales of Text-CNN to process the features extracted by BERT and SWTR, we compare with the following models, and the results are shown in Figs 7 and 8.

- $BERT_{cnn1}$: We use Text-CNN with 64 convolution kernels of size (1,768) to process the text feature $T$ extracted by BERT-Base, and input the processed text features into the classifier to obtain classification results.

- $BERT_{cnn2}$: We use Text-CNN with 64 convolution kernels of size (1,768) and 64 convolution kernels of size (2,768) to process the text feature $T$, and concatenating it input into the classifier to obtain detection results.

- $BERT_{cnn4}$: We use Text-CNN with 64 convolutional kernels of size (1,768), 64 convolutional kernels of size (2,76 8), 64 convolutional kernels of size (3,768) and 64 convolutional kernels

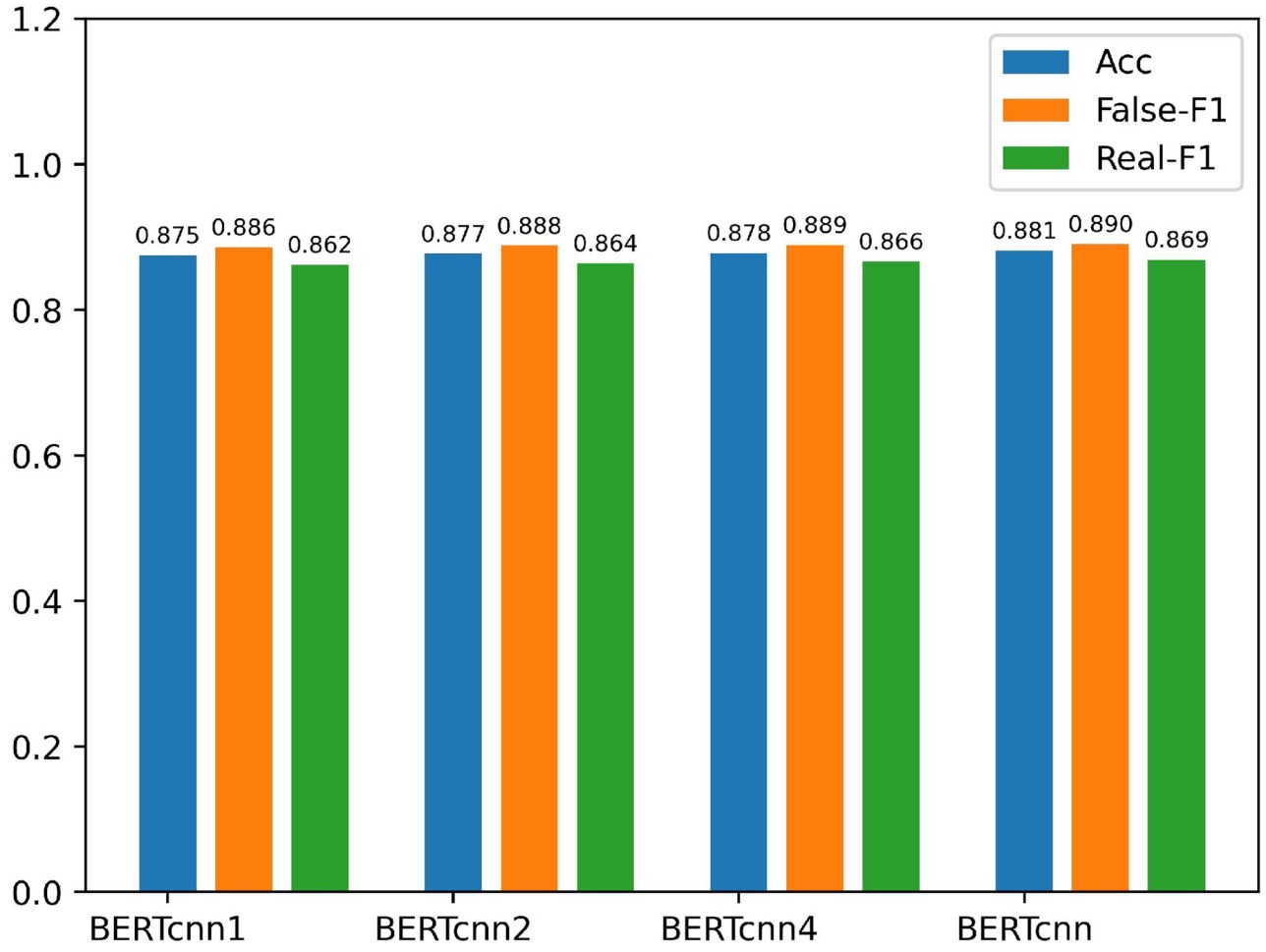

**Fig 7. Comparative results of ablation experiments on the Weibo dataset 3 (1).**

of size (4,768) to process the text features $T$ and concatenate them into the classifier to obtain the results.

- $BERT_{cnn}$: The model used in this paper for detecting the authenticity of posts after processing the text features $T$ using Text-CNN at three different scales.

- $SWTR_{cnn1}$: Similar to the model $BERT_{cnn1}$, except that the processed text features $T$ are replaced with image features $V$.

- $SWTR_{cnn2}$: Similar to the model $SWTR_{cnn2}$, just replace the processed text features $T$ with image features $V$.

- $SWTR_{cnn4}$: Replace the text feature $T$ processed by $SWTR_{cnn4}$ with the image feature $V$.

- $SWTR_{cnn}$: The image features $V_1$, $V_2$ and $V_3$ are concatenated and fed into the classifier to obtain the detection results.

As can be seen in Figs 7 and 8, the accuracy of $BERT_{cnn1}$, $BERT_{cnn2}$, $BERT_{cnn}$ and $SWTR_{cnn1}$, $SWTR_{cnn2}$, $SWTR_{cnn}$ for post detection gradually improves as the number of Text-CNNs of different scales increases. However, when the number of convolution kernels at different scales

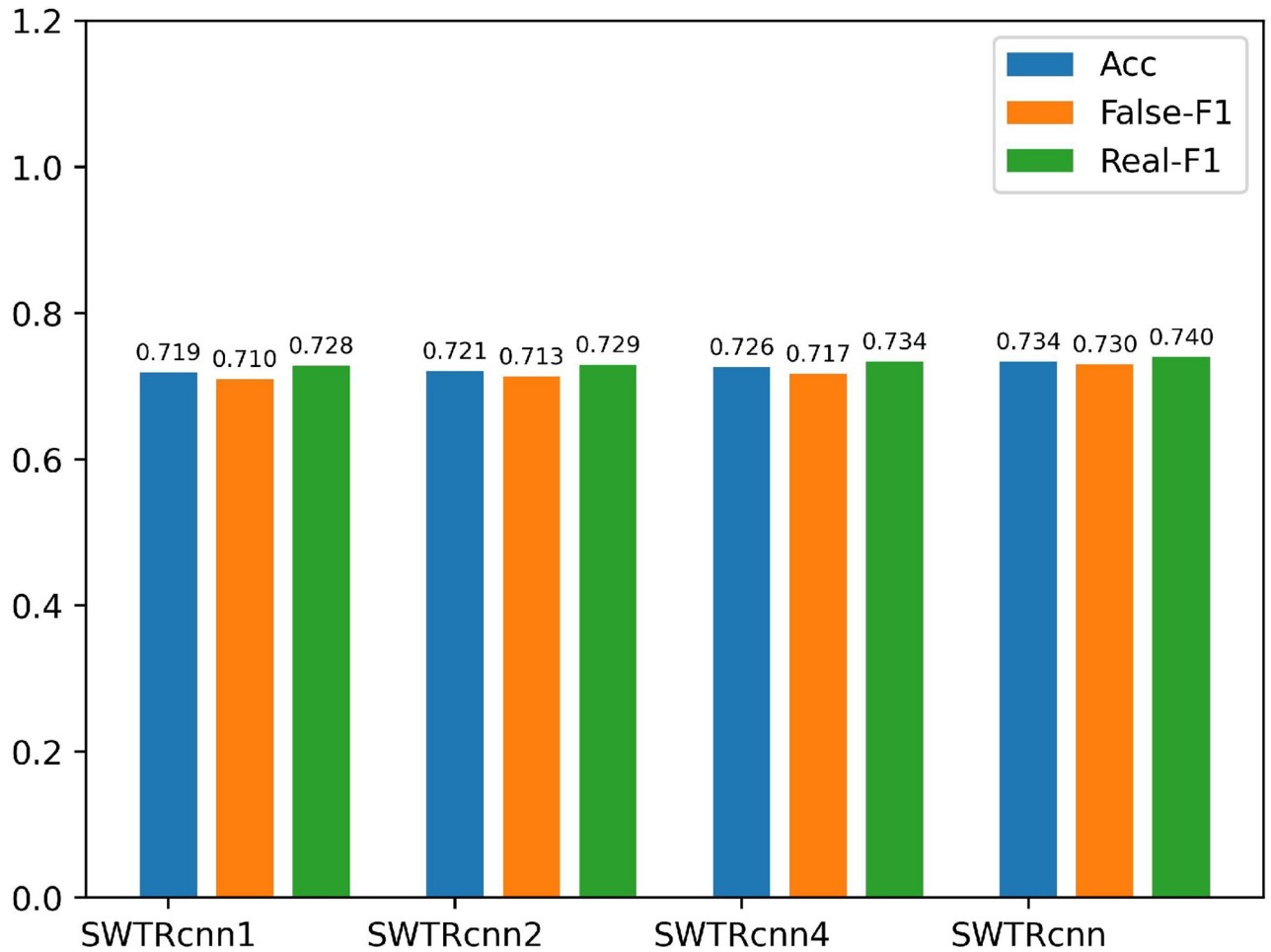

**Fig 8. Comparative results of ablation experiments on the Weibo dataset 3 (2).**

exceeds three. The performance of the model will gradually decrease. Comparing $BERT_{cnn}$, $SWTR_{cnn}$ with $BERT_{cnn4}$, $SWTR_{cnn4}$, the accuracy increases by 0.3% and 0.8%, and the F1 value also increases by 0.3% and 0.6%. The experimental results show that our proposed Text-CNN with three scales (1,768), (2,768) and (3,768) are the most effective to process text features and image features.

We analyze the dataset to further validate our conclusions. We used the jieba word splitter to segment the test dataset from the Weibo dataset and to calculate the number of tokens of different lengths. The distribution is shown in Fig 9.

From Fig 9, we can see that the length of each token in the sentence is not consistent, so when we use Text-CNN of different scales to extract local features similar to n-grams in the text, we will not only extract some valid features, at the same time, some invalid features will be extracted. From Fig 9 we can see that 97% of the tokens in the dataset are less than 4 in length and only 3% of tokens are longer than 3, combined with the experimental results in Fig 7, it can be concluded that when we use Text-CNN with widths of 1, 2, and 3 to extract features, more valid features are extracted than invalid features, thus increasing the performance of the model. When we use larger width Text-CNN, more invalid features than valid features are extracted, thus degrading the performance of the model.

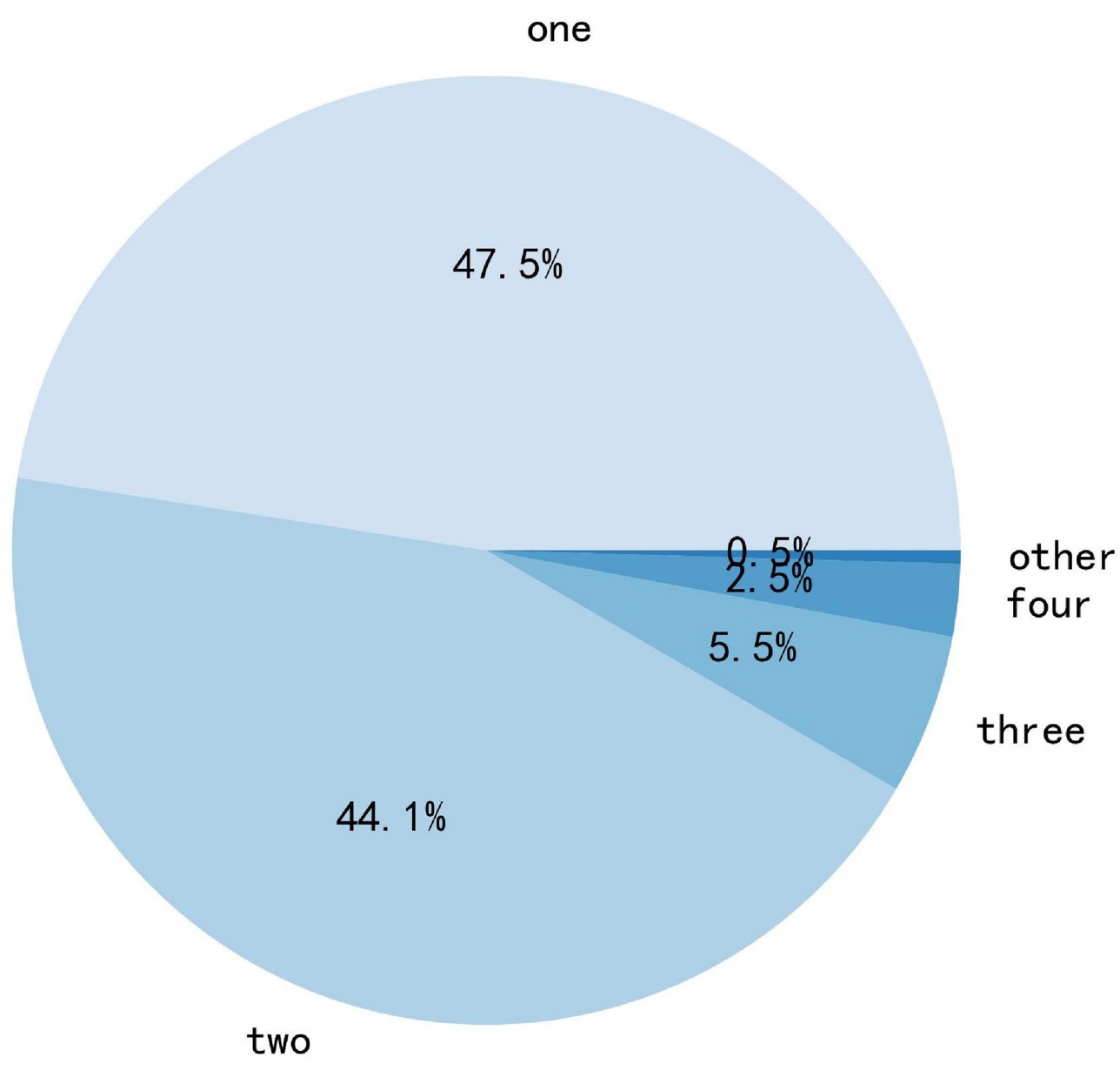

**Fig 9. Frequency of words with different lengths.**

**Ablation experiment four.** To demonstrate the effectiveness of our fusion method, we set up several different fusion models to compare with our fusion model.

- E-Sum (Early Summation): The features of the different modalities are weighted and summed by position and fed into the Transformer for processing.

- E-Con (Early Concatenation): The features of the different modalities are concatenated and fed into the Transformer for processing.

- M-to-O (multi-stream to one-stream): First, two Transformer layers are used to process text features and image features, and then concatenate and input into another Transformer layer for processing.

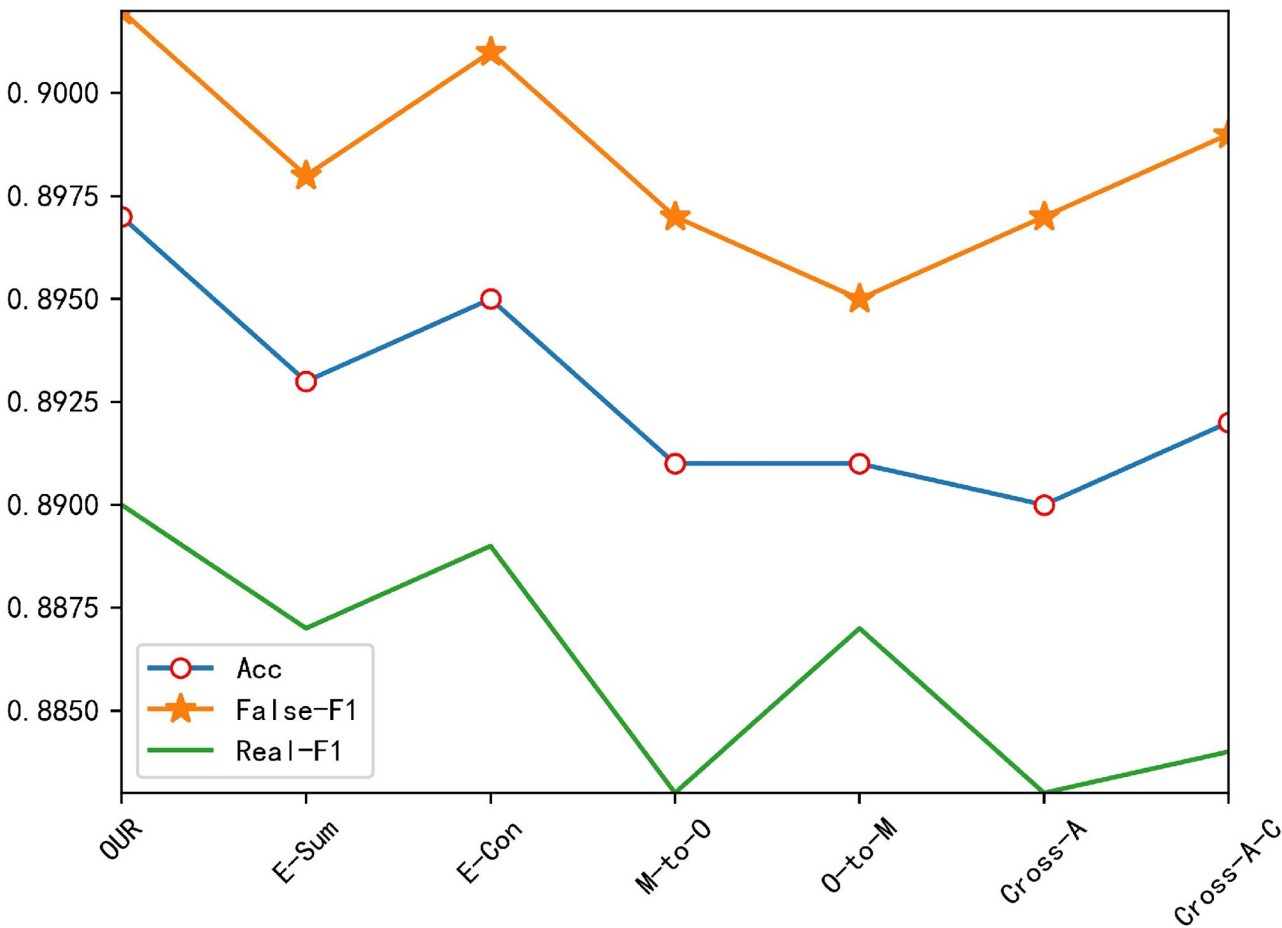

**Fig 10. Experimental results of ablation experiment 4.**

- O-to-M (one-stream to multi-stream): First, the text features and image features are concatenated and input into the Transformer for processing, and then split and input into two Transformers layers for processing.

- Cross-A (Cross-Attention): When using two Transformer layers to process text features and image features, exchange two Q (Query) to complete the fusion of text features and image features.

- Cross-A-C (Cross-Attention to Concatenation): The text features and image features processed by Cross-A are concatenated and input to another Transformer layer for processing.

As can be seen in Fig 10 our model has the best performance, which proves the effectiveness of the feature fusion method we use, and the fusion method we use has a smaller number of parameters than other fusion methods.

As shown in the Table 5, the number of parameters for the fusion methods we used is much smaller than for the rest of the fusion methods.

**Table 5. Comparative results of ablation experiments on the Weibo dataset 1.**

| Model | Nnumber of parameters |
|---|---|
| OUR | 0.70M |
| E-Sum | 11.03M |
| E-Con | 11.03M |
| Mul-to-One | 33.08M |
| One-to- Mul | 33.08M |
| Cross-A | 11.02M |
| Cross-A-C | 22.04M |

## Conclusion

This paper proposes a multimodal false information detection method based on Text-CNN and SE module. The model first uses multi-scale Text-CNN to process text features and image features, and uses the MSE module to fuse multi-modal features to obtain fusion features. Finally, the text features and image features processed by Text-CNN and the fusion feature is simply concatenated as the final fusion feature to detect false information. The comparative experiments demonstrate that our model achieves better results on the Weibo and Twitter datasets than the rest of the models. The ablation experiments validate the effectiveness of our improvements to each module of the model.

In future work, we will mainly study the following issues: (1) How to reduce the size of the model so that it can be deployed on small devices while ensuring detection accuracy. (2) How to extract higher quality features from text and images (3) How to fuse text features and image features more fully.

## Author Contributions

**Conceptualization:** Turdi Tohti.

**Investigation:** Turdi Tohti.

**Methodology:** Yi Liang.

**Project administration:** Askar Hamdulla.

**Software:** Yi Liang.

**Supervision:** Turdi Tohti, Askar Hamdulla.

**Validation:** Yi Liang, Turdi Tohti, Askar Hamdulla.

**Visualization:** Yi Liang.

**Writing – original draft:** Yi Liang.

**Writing – review & editing:** Turdi Tohti, Askar Hamdulla.

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
