## [Decision Letter · Decision Letter 0]

12 Oct 2022

PONE-D-22-26288Multimodal false information detection method based on Text-CNN and SE modulePLOS ONE

Dear Dr. Liang,

Thank you for submitting your manuscript to PLOS ONE. After careful consideration, we feel that it has merit but does not fully meet PLOS ONE’s publication criteria as it currently stands. Therefore, we invite you to submit a revised version of the manuscript that addresses the points raised during the review process.

ACADEMIC EDITOR: I assigned this research paper to two reviewers and received comments with minor revision from both reviewers. Authors has to revise the manuscript as per the reviewer suggestions. 

We look forward to receiving your revised manuscript.

Kind regards,

T. Ganesh Kumar, PhD

Academic Editor

PLOS ONE

Journal Requirements:

Additional Editor Comments:

Authors are requested to update reviewer suggestion in your paper, then resubmit it.

Reviewers' comments:

Reviewer's Responses to Questions

**Comments to the Author**

1. Is the manuscript technically sound, and do the data support the conclusions?

Reviewer #1: Yes

Reviewer #2: Yes

2. Has the statistical analysis been performed appropriately and rigorously? 

Reviewer #1: Yes

Reviewer #2: Yes

3. Have the authors made all data underlying the findings in their manuscript fully available?

Reviewer #1: Yes

Reviewer #2: Yes

4. Is the manuscript presented in an intelligible fashion and written in standard English?

Reviewer #1: Yes

Reviewer #2: Yes

5. Review Comments to the Author

Reviewer #1: Comments 1:

This is an excellent report dealing with significant technical matters. I find no fault whatsoever with the methods, data analysis, or conclusions. The work, as with all work coming from this particular domain, is fundamentally sound. My comments here are concerned solely with the organization of the manuscript. Consideration of these points will, I believe, lead to an improved report that better illustrates the key concepts and conclusions.

Comments 2:

Correction:

Abbreviate the figure 5. 'Iter' in full form

Comments 3:

Suggestion:

For data availability, use Plos one referenced data store from the site due to the shared Github link not working. Only Google Drive works for downloading.

Comments 4:

In line number 369 mentioned, we used the jieba word splitter to segment the test dataset, but in the fig.2 or any of the earlier studies, did not mention this process.

Comments 5:

In table 2 it mentioned image resize into 224,224,3 but in the above paragraph line number 237 stated 'and for the image part, we keep all images the same size'. Repharse the term.

Reviewer #2: Dear Author,

1. Kindly elaborate about your proposed work along with Deep Learning techniques / algorithms in detail.

2. In the dataset the text information contains Chinese language, how you will identify the false news?

3. Refer to the existing model technique, Kindly compare with your proposed work. Detail explanation is required about your implementation work. (Tools, language, flow chart, etc)

4. Twitter dataset analyses and implementation details required. (Webio dataset implementation adequate details provided in manuscript)

5. Add more details about Squeeze-and-Excitation Networks.

6. Kindly mention the total dataset used in each model in table no.3 (Include one more column)

7. Overall the concept is good.

6. PLOS authors have the option to publish the peer review history of their article (what does this mean?). If published, this will include your full peer review and any attached files.

Reviewer #1: **Yes: **Janarthanan Sekar

Reviewer #2: **Yes: **ANANDHAN K

---

## [Author Response · Author response to Decision Letter 0]

17 Oct 2022

Response to Reviewer#1

Point 1: Abbreviate the figure 5. 'Iter' in full form?

Response 1: Thank you for your suggestion. The full form of 'iter' is' Iterations'

Point 2: For data availability, use Plos one referenced data store from the site due to the shared Github link not working. Only Google Drive works for downloading.

Response 2: Thank you for your suggestion, This is our oversight. I have updated the Github link. Github link ‘ https://github.com/MKLab-ITI/image-verification-corpus’

Point 3: In line number 369 mentioned, we used the jieba word splitter to segment the test dataset, but in the fig.2 or any of the earlier studies, did not mention this process..

Response 3: Thank you for your suggestion.

We use the jieba word splitter to facilitate our statistics on the percentage of words of different lengths in the Weibo dataset, as a way to verify the rationality of our text feature processing using convolutional kernels of 3 different scales. In our other experiments, we did not use the jieba word splitter to process the data.

Point 4: In table 2 it mentioned image resize into 224,224,3 but in the above paragraph line number 237 stated 'and for the image part, we keep all images the same size'. Repharse the term.

Response 4: Thank you very much for your valuable comments.

We will change "and for the image part, we keep all images the same size" to "and for the image part,we resize all the images to (224, 224, 3)"

Response to Reviewer#2

Point 1: Kindly elaborate about your proposed work along with Deep Learning techniques / algorithms in detail.

Response 1: Thank you for your suggestion. 

First, we describe our pre-processing process for the data. We resize all the images to (224, 224, 3) and fix the text length (33 for English data and 95 for Chinese data) to be processed in the form of '[CLS]+Text+[SEP]'.

We use the pre-processed data to fine-tune the two pre-trained models, Swin-Transformer and BERT, to make these two models perform better in our task.

Finally, we will introduce our proposed model in detail. Our model first inputs the image with the size of (224,224, 3) into the fine tuned Swin Transformer to obtain the image feature with the size of (49,768), and then inputs the preprocessed text into the fine tuned BERT to obtain the text feature with the size of (33/95,768). Then the image features and text features are processed by (1,768), (2,768) and (3,768) convolution kernels to obtain three (1,64) image features and three (1,64) text features. At the same time, we fuse (49,768) dimensional image features and (33/95,768) dimensional text features to obtain (1, 49) image features and (1, 33/95) text features through our proposed MSE module. Finally, we concatenate three (1,64) and one (1,49) dimensional image features with three (1,64) and one (1,33/95) text features to obtain a joint feature (1,466/528). Finally, we input the (1,466/528) dimensional joint features into a full connection layer with the activation function softmax to obtain the final classification results.

Point 2: In the dataset the text information contains Chinese language, how you will identify the false news?

Response 2: Thank you for your suggestion.

BERT is a powerful text pre-training model that can handle not only English text, but also Chinese text. So for Chinese text we will use the same processing method as English text to discriminate false information, the only difference is the inconsistent length of the text. For the English text, we keep the text length to 33, while the Chinese text length is kept to 95.

Point 3: Refer to the existing model technique, Kindly compare with your proposed work. Detail explanation is required about your implementation work. (Tools, language, flow chart, etc).

Response 3: Thank you for your suggestion. This was an oversight on our part.

Machine configuration and environment for this experiment: CPU: Intel Xeon E5-2630L v3, 62 G memory, 8cores, GPU: NVIDIA GeForce RTX 3090, PyTorch(1.7.1), Python(3.8), Cuda(10.2)

Point 4: Twitter dataset analyses and implementation details required. (Webio dataset implementation adequate details provided in manuscript).

Response 4: Thank you very much for your valuable comments. 

We introduced how to process the Twitter dataset in "Dataset and experimental settings".As to why we do not analyze the Twitter dataset separately, there are several reasons :

1.For the Twitter dataset, the processing is done in a similar way to the Weibo dataset except for the inconsistent text length.

2.The Weibo dataset is more representative than the Twitter dataset. In terms of data volume, the Weibo dataset has more than 30 times more image data than the Twitter dataset. In terms of data composition, the Weibo dataset is closer to real life because most of the posts in the Weiboi dataset are from different events, while most of the events in the Twitter dataset are from the same event. Therefore, we choose the more representative Weibo dataset to do the ablation experiment.

3.Repeating our ablation experiments on the Twitter dataset is not necessary. We set up multiple sets of ablation experiments to validate the effectiveness of our proposed model. To ensure the objectivity of the experimental results, we choose to do all the ablation experiments on the more representative Weibo dataset. The effectiveness of our model has been validated on the more complex Weibo dataset, and we do not consider it necessary to repeat the above ablation experiments on the simpler Twitter dataset.

Point 5: Add more details about Squeeze-and-Excitation Networks.

Response 4: Thank you very much for your valuable comments. We add more details about Squeeze-and-Excitation Networks in the 'Feature fusion' section.The added content is shown below.

The SE module is mostly used for channel feature enhancement of the input feature maps in computer vision tasks. For example, if we input a feature map A with dimensions (H,W,C), the SE module will input A into two full connection layers to get the attention score, and then multiply the feature map A with the attention score in the dimension of the channel to get the final output.

Point 6: Kindly mention the total dataset used in each model in table no.3 (Include one more column).

Response 4: Thank you very much for your valuable comments. 

We mention the datasets used by each model in Table 3. The first column of Table 3 shows the dataset used by the model. As shown in Table 3, the Twitter dataset and the Weibo dataset are followed by the experimental results of seven different models on that dataset.

---

## [Decision Letter · Decision Letter 1]

28 Oct 2022

Multimodal false information detection method based on Text-CNN and SE module

PONE-D-22-26288R1

Dear Dr. Liang,

We’re pleased to inform you that your manuscript has been judged scientifically suitable for publication and will be formally accepted for publication once it meets all outstanding technical requirements.

Kind regards,

T. Ganesh Kumar, PhD

Academic Editor

PLOS ONE

Additional Editor Comments (optional):

Dear Authors,

You have fulfilled reviewer comments.

Reviewers' comments:

Reviewer's Responses to Questions

**Comments to the Author**

1. If the authors have adequately addressed your comments raised in a previous round of review and you feel that this manuscript is now acceptable for publication, you may indicate that here to bypass the “Comments to the Author” section, enter your conflict of interest statement in the “Confidential to Editor” section, and submit your "Accept" recommendation.

Reviewer #1: All comments have been addressed

2. Is the manuscript technically sound, and do the data support the conclusions?

Reviewer #1: Yes

3. Has the statistical analysis been performed appropriately and rigorously? 

Reviewer #1: Yes

4. Have the authors made all data underlying the findings in their manuscript fully available?

Reviewer #1: Yes

5. Is the manuscript presented in an intelligible fashion and written in standard English?

Reviewer #1: Yes

6. Review Comments to the Author

Reviewer #1: (No Response)

7. PLOS authors have the option to publish the peer review history of their article (what does this mean?). If published, this will include your full peer review and any attached files.

---

## [Editor Report · Acceptance letter]

14 Nov 2022

PONE-D-22-26288R1 

Multimodal false information detection method based on Text-CNN and SE module 

Dear Dr. Liang:

I'm pleased to inform you that your manuscript has been deemed suitable for publication in PLOS ONE. Congratulations! Your manuscript is now with our production department. 

Kind regards, 

on behalf of

Dr. T. Ganesh Kumar 

Academic Editor

PLOS ONE